# Exploring the Potential of Linear π-Bridge Structures in a D-π-A Organic Photosensitizer for Improved Open-Circuit Voltage

**DOI:** 10.3390/nano14131106

**Published:** 2024-06-27

**Authors:** Min-Woo Lee, Seunghyun Yoo, Chang Woo Kim

**Affiliations:** 1Department of Chemistry, Sogang University, Seoul 04107, Republic of Korea; yohan-20@hanmail.net; 2R&D Team, The Day1Lab, #1007 Mario Tower, 28 Digital-ro 30-gil, Guro-gu, Seoul 08389, Republic of Korea; shyoo@day1-lab.com; 3Department of Nanotechnology Engineering, College of Engineering, Pukyong National University, Busan 48513, Republic of Korea

**Keywords:** donor–acceptor photosensitizer, electron-rich unit, molecular structure, dye-sensitized solar cells, impedance spectroscopy

## Abstract

We present the design, synthesis, and evaluation of novel metal-free photosensitizers based on D-π-A structures featuring tri-arylamine as an electron donor, cyanoacrylic acid as an anchoring group, and substituted derivative π-bridges including 9,9-dimethyl-9H-fluorene, benzo[*b*]thiophene, or naphtho [1,2-*b*:4,3-*b*′]dithiophene. The aim of the current research is to unravel the relationship between chemical structure and photovoltaic performance in solar cell applications by investigating the properties of these organic sensitizers. The newly developed photosensitizers displayed variations in HOMO-LUMO energy gaps and photovoltaic performances due to their distinct π-bridge structures and exhibited diverse spectral responses ranging from 343 to 490 nm. The t-shaped and short linear photosensitizers demonstrated interesting behaviors in dye-sensitized solar cells, such as the effect of the molecular size in electron recombination. The study showed that a t-shaped photosensitizer with a bulky structure reduced electron recombination, while short linear photosensitizers with a smaller molecular size resulted in a higher open-circuit voltage value and enhanced photovoltaic performance. Impedance analysis further supported the findings, highlighting the influence of dye loading and I_3_^−^ ion surface passivation on the overall performance of solar cells. The molecular design methodology proposed in this study enables promising photovoltaic performance in solar cells, addressing the demand for highly efficient, metal-free organic photosensitizers.

## 1. Introduction

Solar energy has undoubtedly become a pivotal source of clean, renewable energy in the modern world [1,2,3]. As global demand for sustainable energy solutions soars, researchers are ardently delving into innovative materials and groundbreaking concepts to amplify solar energy’s conversion efficiency [4,5]. A critical component in this pursuit is the photosensitizer, a key player in capturing and converting sunlight into electricity [6,7,8]. The domain of photosensitizers is flourishing with pioneering research, unlocking new avenues for the development of cost-effective, environmentally friendly, and high-performing solar energy-harvesting technologies [9,10,11]. The implementation of photosensitizers in solar cell technology has garnered immense interest due to their notable potential in enhancing performance, efficiency, and cost-effectiveness [12,13]. Nevertheless, challenges and trade-offs persist in their incorporation into solar energy-harvesting systems [14,15].

Ruthenium-based photosensitizers, particularly ruthenium polypyridyl complexes, have been the subject of extensive research due to their first-rate light-harvesting properties and favorable redox characteristics in photovoltaics [16,17]. Highly rated ruthenium organic photosensitizers such as N3, N719 (black dye), N749 (red dye), and Z907 each offer unique attributes that contribute to their widespread utilization [18,19]. N3, one of the earliest and most comprehensively investigated organic dyes, is characterized by its potent visible light absorption, elevated molar extinction coefficient, and outstanding electron injection efficiency [20]. N719 is known for its superior performance and stability, while N749 features an extended absorption range in the near-infrared (NIR) region, enabling the harvesting of a more expansive solar spectrum [21,22]. Finally, Z907, as an amphiphilic dye, provides enhanced stability and optimal molecular organization within photovoltaic solar cells [23]. Despite their remarkable performance, concerns surrounding the cost, availability, and environmental impact of noble metal-based organic photosensitizer-like ruthenium complexes have driven a search for alternative solutions [17].

Donor-π-acceptor (D-π-A) photosensitizing dyes emerge as an enthralling alternative, boasting versatility, affordability, and customizable design options [24,25,26]. The essence of D-π-A organic dyes lies in comprehending and perfecting their structural properties to maximize solar cell performance, particularly by integrating linear π-bridge structures [27]. As metal-free organic photosensitizers, variable structures such as Cyanine dyes, Merocyanine dyes, Coumarin dyes, Indoline dyes, Hemicyanine dyes, Triphenylamine, and Xanthene dyes have been suggested for photovoltaic applications [28]. Despite their promising aspects, challenging issues remain, such as their low device efficiency compared to traditional metal-based photosensitizers, their long-term stability in harsh operating conditions, wider and stronger absorption spectra, and the optimization of D-π-A systems in metal-free organic dyes [11,12,28].

In this research, we investigate the correlation between the chemical structure and photovoltaic performance of organic photosensitizers, with a focus on D-π-A dyes and linear π-bridge structures. This study introduces novel conjugated organic molecules with systematically varied chemical structures, elucidating their photovoltaic properties. The current work provides comprehensive data on their chemical structures, UV-Vis absorption spectra, cyclic voltammograms, HOMO and LUMO energy levels, and other relevant properties. Our research examines the role of D-π-A structures and the impact of linear π-bridges on the performance of organic photosensitizers in solar cell technology. This investigation aims to identify the benefits and challenges of D-π-A organic photosensitizers, contributing to the development of efficient, affordable, and sustainable solar energy solutions.

## 2. Materials and Methods

### 2.1. Synthesis of Organic D-π-A Photosensitizer

All reactions were conducted under a nitrogen atmosphere using flame-dried and oven-dried glassware with magnetic stirring. Air-sensitive reagents and solutions were transferred via syringe or cannula and were introduced to the apparatus through rubber septa. Tetrahydrofuran (THF), diethyl ether, benzene, and toluene were distilled from sodium/benzophenone. Dichloromethane (DCM) was distilled from phosphorous pentoxide. Acetonitrile, 1,2-dimethoxyethane (DME) and N,N-dimethylformamide (DMF), dimethyl sulfoxide (DMSO), triethylamine (TEA), and other nitrogen-containing bases were simple- or vacuum-distilled from calcium hydride. Products from reactions were characterized by thin layer chromatography (TLC) with 0.25 mm plates (E. Merck pre-coated silica gel plates, 60 F254). Visualization was achieved either using ultraviolet (UV) light or by immersing the samples in solutions of p-anisaldehyde or phosphomolybdic acid (PMA), followed by heating on a hot plate for 15 s. Purification of reaction products was carried out by flash chromatography using an EM reagent silica gel 60 (230–400 mesh). ^1^H-NMR and ^13^C-NMR spectra were obtained using a Varian Gemini-300 (300 MHz for ^1^H, and 75 MHz for ^13^C) and a Varian Inova-500 (500 MHz for ^1^H, and 125 MHz for ^13^C) spectrometer. Chemical shifts are reported relative to chloroform (δ 7.26) for ^1^H NMR and chloroform (δ 77.2) for ^13^C NMR. Data are recorded as coupling constant(s) in Hz (br = broad, s = singlet, d = doublet, t = triplet, q = quartet, m = multiplet). The IR spectra were recorded on a Mattson galaxy 2020 FT-IR spectrometer. Elemental analyses were performed by the Organic Chemistry Research Center (OCRC) at Sogang University (Seoul, Republic of Korea) using a Carlo Erba EA 1180 elemental analyzer. LC/Mass was recorded on a Waters Auto Purification System LC/MS. High-resolution mass spectrum was recorded on a 4.7 Tesla IonSpec ESI-FTMS or a Micromass LCT ESI-TOF mass spectrometer. All commercially available compounds were used as received unless stated otherwise.

### 2.2. Fabrication of Solar Cell Devices

Screen-printable pastes composed of nanocrystalline TiO_2_ particles (~20 nm) and large TiO_2_ particles (CCIC, 400 nm, Tokyo, Japan) were prepared by following the reported procedure [27]. Nanocrystalline TiO_2_ film was deposited on fluorine-doped tin oxide (FTO) glass (Pilkington, TEC-8, 8 Ω/sq, 2.3 mm thick) that was pre-coated with Ti(IV) bis(ethyl acetoacetato)-diisopropoxide solution, which was heated at 500 °C for 30 min at a heating rate of 5 °C/min. For the bi-layer structure, the CCIC TiO_2_ particle layer was deposited on the annealed nanocrystalline TiO_2_ films and heated at 500 °C for 30 min. In the typical fabrication of dye-sensitized solar cells, the resulting TiO_2_ double-layered films (8.2 + 6.8 μm) were immersed in anhydrous ethanol containing 0.5 mM of synthesized photosensitizing dye and kept for 16 hrs at an ambient temperature. As counter-electrodes, Pt electrodes were prepared on the FTO glasses by using 0.7 mM of H_2_PtCl_6_ solution, followed by heating at 400 °C for 20 min in air. In the sealed cell, the electrolyte solution contained 0.6 M 1-methyl-3-propyl imidazolium iodide (PMII), 0.1 M LiI, 0.03 M I_2_, and 0.5 M 4-tert-butylpyridine (TBP) dissolved in acetonitrile. Active areas of photosensitizing dye-coated TiO_2_ films were in the range of 0.45 ± 0.3 cm^2^, which was measured using an image analysis program equipped with a CCD camera (Moticam 1000, Hong Kong). TiO_2_ film thickness was measured using an α-step surface profiler (KLA tencor, Milpitas, CA, USA).

### 2.3. Performance Measurements

UV-VIS spectra of the synthesized photosensitizer were recorded by using Agilent 8453 (Santa Clara, CA, USA). The ethanol-dispersed photosensitizing dye solution was transferred into a quartz cell with a 1 cm path length to measure signals. Photocurrent–voltage (I–V) measurements were performed using a Keithly model 2400 source measure unit (Beaverton, OR, USA). An Oriel solar simulator (Irvine, CA, USA) equipped with a 1000 W Xenon lamp was used as a light source. The light intensity was adjusted by using an NREL-calibrated Si solar cell equipped with a KG-5 filter to simulate 1 sun light intensity. The photocurrent–voltage measurement of solar cells was performed with an aperture mask [27]. Incident photon-to-current conversion efficiency (IPCE) was measured as a function of wavelength from 300 to 800 nm by using a specially designed IPCE system manufactured by PV measurements, Inc. (Point Roberts, WA, USA). A 75 W Xenon lamp was used as a light source to generate monochromatic beam. Calibration was performed by using a silicon photodiode, which was calibrated using the NIST-calibrated photodiode G425. IPCE values were collected under biased white light at a low chopping speed of 10 Hz.

## 3. Results and Discussion

### 3.1. Synthesis Route of Organic Photosensitizers and Their Chemical Structures

Here, we suggest four new kinds of metal-free organic photosensitizers to address the current issues presented in Figure 1. A newly designed metal-free organic photosensitizer features a straightforward push–pull structure, which comprises an electron donor (4-(bis(9,9-dimethyl-9H-fluoren-2-yl)amino, π-bridge, and α-cyanoacrylic acid as an electron acceptor/anchor. The objective of the current design is to investigate the impact of the π-bridge on the photosensitizer’s performance. The introduction of four distinct π-bridges is anticipated to have a significant influence on the optical and electronic properties of each individual molecular structure, which in turn may result in varying power conversion efficiencies. Detailed information on the chemical structures and synthetic routes of the novel organic photosensitizer series (PS) is shown in the Appendix A. The purpose of the designed PS is to assess the impact of the π-bridge on achieving a higher open-circuit voltage. The four newly developed organic photosensitizers, denoted as shorter linear (_sh_PS), longer linear (_lo_PS), c-shaped (_c_PS), and t-shaped (_t_PS), were synthesized to scrutinize factors impacting the electrochemical performances of solar cells. As the shortest photosensitizer, _sh_PS is anticipated to exhibit the highest packing factor. This characteristic suggests that it should achieve maximal adsorption on the TiO_2_ surface within a solar cell. Isomer organic photosensitizing dyes, _lo_PS and _c_PS, which have unique steric effects, were employed to examine their electron recombination rates and open-circuit voltages. Additionally, the bulkiest photosensitizer, _t_PS, is expected to prevent interactions between the I_3_^−^ ion and the TiO_2_ surface in the solar cell. This was introduced specifically for analyzing short-circuit currents [29].

Starting from fluorene, iodination was carried out by continuously refluxing iodine and periodic acid in a dissolved AcOH-H_2_SO_4_ solution, as shown in Appendix A. The iodide underwent dimethylation in step b, reacting with CH_3_I and KOH. The yields for the iodide and dimethyl derivatives were 51% and 91%, respectively [30]. In step c, nitration of the dimethyl derivative was achieved by reacting with HNO_3_-AcOH at 80 °C, yielding 81%. Step d involved a copper-catalyzed displacement of the nitrated substrate, refluxing with copper(I) cyanide overnight and yielding 99% [31]. The cyanate substrate was reduced in step e with a yield of 88%, using SnCl_2_ to introduce an amine functional group to the substrate [32]. Aryl substitution in step f of the amine substrate was conducted with a 68% yield through copper-catalyzed amination [30]. Progressing from product 6 to product 7 in step g, the substrate underwent reduction with a 78% yield using DIBAL [33]. The resulting aldehyde (product 7 from step h, SI) underwent condensation with cyanoacetic acid in the presence of piperidine to form _sh_PS, with a 48% yield. Finally, the metal-free organic photosensitizer, _sh_PS, was successfully synthesized through the reactions outlined in Figure 1 [34].

To synthesize _c_PS and _lo_PS, a series of three-step reactions was carried out. As shown in Appendix A, the S_N_2 substitution reaction of bromoacetal with 3-bromobenzenethiol was completed with a 100% yield, converting product 8 to product 9 under a THF solvent environment through continuous reflux [35]. The deprotection and cyclization of product 9, catalyzed by polyphosphoric acid (PPA), resulted in an inseparable 1:1 mixture of product 10 and product 11 with a 36% yield. The final formylation of the mixture enabled the separation of bromoaldehyde, product 12 and product 13, with a combined yield of 60% [36]. Product 16 was prepared using copper-catalyzed displacement of iodide 2 by 4-bromoaniline (product 14), obtaining a 42% yield. This was followed by a metal–bromide exchange and the addition of 2-isopropoxy-4,4,5,5-tetramethyl-1,3,2-dioxaborolane, yielding 73%, as shown in Appendix A [30]. Subsequently, the condensation of aldehyde 18 with cyanoacetic acid in the presence of piperidine (95%) produced the metal-free organic photosensitizer, _c_PS, as shown in Appendix A [34]. The metal-free organic photosensitizer, _lo_PS, was synthesized through palladium-catalyzed Suzuki coupling of bromide (product 17) with dioxaborolane 16, with a 56% yield, as shown in Appendix A.

The synthesis of the metal-free organic photosensitizer, _t_PS, requires bromoaldehyde 49, as shown in Appendix A. Initially, 1-bromonaphthalene was brominated selectively at the C-4 position using Br_2_, yielding 85% [37]. The 1,4-dibromo substituents of product 44 were entirely replaced by the methylthio group, resulting in sulfide 45 with a 53% yield [38]. Reductive demethylation of 45 using Na in DMA (60%) and the S_N_2 substitution reaction of the bromoacetal by aryl thiol groups produced intermediate 46 (60%) [35]. PPA-catalyzed deprotection and cyclization of intermediate 46 occurred with an 88% yield, followed by selective formylation of 47 (80%) and the final bromination of 48, resulting in intermediate 49 [36,39]. The metal-free organic photosensitizer, _t_PS, was synthesized by palladium-catalyzed Suzuki coupling of bromide 49 with dioxaborolane 16 (67%). The condensation of aldehyde 50 with cyanoacetic acid in the presence of piperidine (59%) produced the organic photosensitizer _t_PS [37]. The chemical structures of the suggested organic dyes were characterized using ^1^H NMR, ^13^C NMR, and mass spectroscopy.

### 3.2. Optical Properties of Metal-Free Organic Photosensitizer

The light absorption properties of the newly synthesized organic photosensitizers provide valuable information on solar cells’ performance. The molar extinction coefficient, or light absorption coefficient, is an essential indicator of light harvesting efficiency. Generally, a higher molar extinction coefficient value leads to increased light harvesting efficiency. In Figure 1a, the observed shoulder peaks from the UV-Vis spectra are at 370 and 426 nm for _sh_PS and at 350 and 412 nm for _c_PS. These peaks indicate direct evidence of π-π* intramolecular charge transfer [40]. The energy level of the lowest optically active excited states, the π* orbital of the organic photosensitizer, is electronically related to the highest unoccupied state of TiO_2_. Thus, the shoulder peaks provide crucial information regarding the electron injection into the system. According to the Frenkel exciton model, a bound electron–hole pair does not easily dissociate into free carriers. Since the photosensitizer exciton’s binding energy is larger than the thermal energy at room temperature (*k*_B_T ≈ 0.025 eV), the exciton must migrate to the heterojunction and dissociate to generate a current [41,42].

The computed extinction coefficients at peak wavelengths and the electrochemical properties of the synthesized organic sensitizers are compared in Table 1. Remarkably high molar extinction coefficients were observed for all organic molecules (ε > 5 × 10^5^ M^−1^ cm^−1^), suggesting that exciton generation will be promoted in solar cell devices. It can be noted that the conjugation length is strongly dependent on the π-bridge structures. Among the organic sensitizers, _sh_PS displayed the largest amount of photosensitizers loaded on TiO_2_, as shown in Table 1. This finding implies that the linear molecular structures are correlated with a longer conjugation length. The molecular structure of _sh_PS highlights the significance of a well-developed conjugation network for improved solar cell performance. Due to the absence of a planar thiophene ring in _sh_PS, its extinction coefficient is smaller than those of other organic photosensitizers, resulting in a decreased light absorption range. Nevertheless, the compact and well-developed conjugation network of _sh_PS enables better charge transfer, leading to an enhanced conversion efficiency. In contrast, _t_PS exhibits a narrow absorption range owing to the presence of additional fused rings. The naphthodithiophene structure deteriorates the π-conjugation network and the push–pull effect, resulting in the lowest conversion efficiency among the studied organic photosensitizers [43].

E_0-0_ energies, representing HOMO-LUMO transitions, were estimated from the onset wavelength of UV-Vis absorption (λ_onset_). The equation for the HOMO-LUMO transition energy gap is given in the following Equation (1) [44]:(1)$$E_{0-0}=\frac{1240}{\lambda_{onset}}\;\left(\mathrm{nm}\right)$$

Additionally, the π-conjugation length is significantly influenced by the structural factor between the donor and the π-bridge. For example, the E_0-0_ values follow the order of 4-phenylbenzothiophene (E_0-0_ = 2.61 eV) = naphthodithiophene (E_0-0_ = 2.61 eV) < 6-phenylbenzothiophene (E_0-0_ = 2.44 eV) < 9,9-dimethyl-florene (E_0-0_ = 2.43 eV). To assess the electrochemical properties depicted in Figure 1b, cyclic voltammetry measurements were conducted on four kinds of synthesized organic photosensitizers in a THF solvent environment. A 0.1 M concentration of tetrabutylammonium tetrafluoroborate was added as a supporting electrolyte. Using a CHI 600C electrochemical analyzer, Pt wires were employed as both the working and counter-electrodes, while an Ag wire served as the pseudo-reference electrode. The voltage sweep rate was set at 50 mV/s. Oxidation potentials (E_ox_ vs. NHE) were calculated by adding 0.63 V to the oxidation potential of ferrocene (E_ox_ = 0.81 V vs. Ag/Ag^+^) [45]. The first oxidation potential (E_ox_^1^), indicative of the HOMO energy levels in the four kinds of organic sensitizers, was observed within the potential range of 0.81 to 0.87 V, as shown in Table 1. The appearance of the initial oxidation hump is closely related to the oxidation of the 4-(bis(9,9-dimethyl-9H-fluoren-2-yl)amino terminal [46]. Interestingly, the impact of individual π-bridge structures was evident from the potential values of the secondary oxidation (E_ox_^2^) in the potential range of 1.09 to 1.20 V. Remarkably, π-bridge structures play a more vital role in determining LUMO levels than they do in HOMO levels. This observation suggests that each organic dye possesses sufficiently higher E_LUMO_ to maintain the minimum energy difference of 0.4 eV in relation to TiO_2_ (−0.28 V vs. NHE) of the conduction band potential (E_cb_). This difference is essential for efficient electron injection from a sensitizer to the conduction band of TiO_2_. Considering that all oxidation potentials are more positive than the iodide/triiodide couple (0.4 V vs. NHE), favorable dye regeneration kinetics can be anticipated. Considering that LUMO energy levels and E_0-0_* values were approximated by the energy difference between E_ox_^1^ and E_0-0_, the E_0-0_* values ranged from −1.57 to −1.80 V versus NHE. Consequently, efficient electron injection from the photoexcited sensitizer to the TiO_2_ conduction band (−0.4 V vs. NHE) is expected. These empirical findings demonstrate that both the HOMO and LUMO energy levels are well aligned for effective charge separation and dye regeneration.

### 3.3. Quantum Mechanical Calculations of Photoexcitation Phenomenon

To investigate the relationship between the geometric structure of the four metal-free organic photosensitizers and their electrochemical properties, quantum mechanical analysis based on DFT calculations was performed using the B3LYP/6-31G* level of theory. The electron density surfaces of HOMO and LUMO for frontier molecular orbitals were calculated for each of the four organic sensitizers, as shown in Figure 2 [47]. The HOMO and LUMO orbitals of PS are uniformly delocalized from the donor to the acceptor, resulting in favorable conditions for efficient electron transfer. Each organic sensitizer exhibits a distinct dihedral angle between 9,9-dimethyl-9H-fluorene, benzo[b]thiophene, and naphtho[1,2-b:4,3-b′]dithiophene as π-bridge units and the adjacent donor unit. _lo_PS at 343 nm and _t_PS at 370 nm exhibit more extended π-conjugation than _sh_PS at 426 nm. Intriguingly, given that the dihedral angles of _lo_PS and _t_PS are larger than 30°, it appears that the π-conjugation effect is hindered by steric factors due to the bulky structure and the structural connection between the donor and π-bridge units. This behavior in the DFT calculations is consistent with the information derived from the UV-vis spectra. The differences in dihedral angles, along with optical and physical properties, are expected to influence a solar cell’s performance.

### 3.4. Photovoltaic Performances

Dye-sensitized solar cells were fabricated following the described procedures, and their photovoltaic performances were measured under an AM 1.5 global one sun light intensity of 100 mW/cm^2^. The incident photon-to-current conversion efficiencies (IPCEs) of the four organic photosensitizers were plotted as a function of the incident wavelength, as shown in Figure 3a. The maximum IPCE values for each sensitizer were observed in the range of 55 to 82%. The onset wavelengths were recorded between 620 nm and 624 nm, following the order of _sh_PS > _c_PS > _lo_PS > _t_PS. The light absorption range of _t_PS is broader than that of _lo_PS due to its bulky structure, resulting in a smaller amount of adsorbed _t_PS on the TiO_2_ surface compared to _lo_PS. Consequently, the IPCE value of _t_PS was the lowest among the four organic sensitizers. Upon comparing the IPCE spectra and the UV-vis spectra, all organic PSs exhibited a red-shift of approximately 100 nm [45]. This tendency extends the light absorption range by nearly 100 nm. Notably, _sh_PS and _c_PS, with linear π-bridge structures, demonstrate high photon-to-current conversion efficiencies exceeding 70% due to their effective conjugation.

Figure 3b and Table 2 display the short-circuit photocurrent density (Jsc) as a function of voltage, as well as the open-circuit voltage (Voc), fill factor (FF), and overall power conversion efficiency (η). The repeated performances of the four typical samples are summarized in Appendix A. The Jsc values demonstrated an increasing trend in the order of _t_PS < _lo_PS < _c_PS < _sh_PS, which corresponds to their tendencies in the IPCE data. While the Voc values for _lo_PS and _c_PS were observed to be between 735 and 745 mV, _sh_PS and _t_PS exhibited higher Voc values of above 830 mV. Remarkably, the Voc values of _sh_PS and _t_PS were noticeably higher than those of typical metal-free organic sensitizers reported in the literature. The origin of this behavior is still under investigation, but π-bridges of 9,9-dimethylflorene and naphtho[1,2-b:4,3-b′]dithiophene are hypothesized to promote Voc in other kinds of donor systems. _lo_PS and _t_PS, as isomers with varying degrees of linearity, exhibited a discrepancy in conversion efficiencies, as presented in Table 2. A higher conversion efficiency was recorded for _lo_PS, which featured a more linear molecular structure. Although _t_PS exhibited a lower adsorption capacity compared to _lo_PS, its relatively large molecular size appeared to significantly impede the approach of I_3_^−^ to TiO_2_ surface. _lo_PS and _t_PS, with bent π-bridge structures, displayed lower FF values of 0.66 compared to other organic sensitizers. The overall power conversion efficiencies for the four typical photosensitizers were observed within a range of 2.90% to 4.48%. The wide-ranging distribution of conversion efficiencies with the same donor and acceptor units underscores the significance of the π-bridge structure for achieving high performance.

### 3.5. Analysis of Electrochemical Impedance Spectroscopy

The photovoltaic properties of solar cells employing the PS series were further investigated using electrochemical impedance spectroscopy (EIS) analysis. Figure 4 presents the fitting parameters of solar cells with PSs under dark conditions at bias potentials ranging from −0.3 to −0.6 V, obtained using ZView software. The equivalent circuit model comprises series resistance (R_s_) and the impedance at the electrolyte/Pt interface (R_Pt_ and CPE_1_) and the electrolyte/TiO_2_/dye interface (R_ct_ and CPE_2_), as shown in Appendix A [36]. Among the photosensitizing dyes, _sh_PS demonstrated the highest interfacial charge resistance (R_ct_). Since the surface passivation of I_3_^−^ ions in the electrolyte depends on the amount of adsorbed dye on the TiO_2_ film, _sh_PS minimizes the exposure of the TiO_2_ surface, thereby reducing electron recombination. Considering only linear π-bridge structures, the R_ct_ values decreased in the order of _sh_PS to _c_PS, aligning with the trend observed in dye loading amounts presented in Table 1. CPE_2_ provides information about the chemical capacitance (C_μ_) of TiO_2_, which offers an evaluation of the total electron density. The C_μ_ value serves as an indicator of the number of photo-injected electrons accumulated in the TiO_2_ electrode, as depicted in Figure 4a. The number of photo-injected electrons is heavily influenced by electron recombination with electrolytes; hence, the number of electrons in the TiO_2_ electrode significantly decreases as the electron recombination increases. Although _t_PS had a low R_ct_ value, it exhibited a higher C_μ_ than _sh_PS due to the enhanced dipole moment at the TiO_2_ surface, as shown in Figure 4b. The C_μ_ values followed the order of _lo_PS < _c_PS < _sh_PS < _t_PS. Consequently, the electron lifetime (τ_n_), calculated as a product of R_ct_ and C_μ_, decreased in the order of _sh_PS > _t_PS > _c_PS > _lo_PS, as portrayed in Figure 4c. This trend matched the one listed in Table 2. The electrochemical impedance analyses elucidated the chemical and structural features of the PS series, which were directly related to the electrochemical and photovoltaic properties of photoanodes.

## 4. Conclusions

In conclusion, this study presents the design and synthesis of four novel metal-free photosensitizers featuring D-π-A structures, utilizing triarylamine as an electron donor, cyanoacrylic acid as an anchoring group, and π-bridges substituted with 9,9-dimethyl-9H-fluorene, benzo[b]thiophene, or naphtho[1,2-b:4,3-b′]dithiophene derivatives. These dyes were developed to investigate the relationship between the chemical structure and photovoltaic performance in solar cell applications. The PS series of photosensitizers showed variations in HOMO-LUMO energy gaps and photovoltaic performances due to differences in their π-bridge structures, with spectral responses ranging from 343 to 490 nm. The T-shaped photosensitizer (_t_PS) demonstrated a bulky structure that reduced the electron recombination rate but limited dye loading on the TiO_2_ surface. Conversely, the shorter linear photosensitizer (_sh_PS) exhibited the highest open-circuit voltage due to its smaller molecular size, which facilitated increased dye adsorption and resulted in a longer electron lifetime within the photoanode. Impedance analysis revealed a significant decrease in the electron recombination rate due to surface passivation by _sh_PS molecules on the TiO_2_ surface. While the _t_PS sensitizers had a narrow absorption band, their linear π-bridges improved dye loading and I_3_^−^ ion surface passivation in the electrolyte, leading to enhanced solar cell efficiency. This study’s molecular design methodology for organic photosensitizers demonstrates promising photovoltaic performance, addressing the demand for highly efficient solar energy devices.

## Data Availability

The produced data are available from the authors upon reasonable request.

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
