# Peer review of "Exploring the Potential of Linear π-Bridge Structures in a D-π-A Organic Photosensitizer for Improved Open-Circuit Voltage"

_nanomaterials, 2024, doi:10.3390/nano14131106_

Round 1

Reviewer 1 Report

Comments and Suggestions for Authors

This manuscript titled with "Exploring the Potential of Linear π-Bridge Structures in D-π-A Organic Photosensitizer for Improved Open-Circuit Voltage" from Lee et al. presents the design, synthesis, and evaluation of novel metal-free photosensitizers based on D-π-A structures featuring tri-arylamine as an electron donor, cyanoacrylic acid as an anchoring group, and substituted derivative π-bridges including 9,9-dimethyl-9H-fluorene, benzo[b]thiophene, or naphtho[1,2-b:4,3-b']dithiophene. The study showed that t- shaped photosensitizer reduced electron recombination, and short linear photosensitizers improved the open-circuit voltage. This finding is interesting for unravelling the relationship between chemical structure and photovoltaic performance in solar cells. Overall, the manuscript can meet the requirements of Nanomaterials provided that the following issues are properly addressed:

1.       The raw data of impedance analysis should be provided to reflect a more complete electrical properties of the devices.

2.       The NMR spectra of different molecules should also be provided.

Author Response

  1. The raw data of impedance analysis should be provided to reflect a more complete electrical properties of the devices.

We thank reviewer for the valuable suggestion. Regarding electrochemical characterization, current research team have published many research articles till date and we recognized that raw data does uploaded to main draft and SI for publication. We are very sorry for this issue. However, we would like to suggest EIS curve fitting data to SI as an additional result.    

  1. The NMR spectra of different molecules should also be provided

We thank reviewer for the valuable suggestion. We have tried to get a clean NMR data regarding your requests till today but we are very sorry to say that what we could get for NMR is the number of peaks. We concluded that photograph from old NMR is not proper for publication. We hope editor and reviewers understand this issue.    

Reviewer 2 Report

Comments and Suggestions for Authors

1       A brief summary:

The presented work concerns the most topical area in modern applied science: the development of efficient and environmentally friendly energy sources. The field of photovoltaics, which involves the creation of dye-sensitized solar cells, emerged several decades ago and has been continuously developing since then. The scientific community is increasingly engaged in the pursuit of an alternative to silicon solar cells that is worthy of consideration. Consequently, the topic and direction of this research are evident. Synthetic chemists, in a manner analogous to architects, create intricate chemical structures that evoke a sense of wonder. Dyes can be classified as metal-based organic complexes or metal-free organic compounds. In this instance, the structure of the compound is distinguished by the presence of a donor, acceptor, and π-linker. The sequence and quantity of the connections between these components determine the existence of several types of structural configurations. This manuscript describes the synthesis and study of key photo-physical properties of metal-free organic dyes of the D-π-A type, as well as the characteristics of solar cells based on them. The authors successfully synthesized five novel compounds in which the role of donor was fulfilled by triethylamine and cyanoacrylic acid acted as an anchoring group. The structure of the pi-linker was varied in order to elucidate the structure-performance relationship in the investigated series of dyes.

2       Questions, remarks and recommendations:

2.1  Abstract. Line 17. The word "unique" is not entirely appropriate in this context due to the fact that the properties of the compounds presented in this paper are not unique, but rather typical of them.

2.2  Introduction. The last paragraph. The description of the research in this paragraph is presented in a somewhat artistic style with excessive pathos about the results of the work. This impression arises from the use of the words "enthralling", "fascinating narrative", "shedding light" "the merits and challenges", "illuminating the ongoing search". It is not my intention to belittle the significance of your work. However, it is important to note that only five compounds are described in this study. Given the vast amount of information published today, this is an important study. Nevertheless, it is but one of many works aimed at understanding the complexity of donor-acceptor systems and their properties. In other words, the structure-property relationship observed in a series of five compounds is unlikely to be a significant contributor to the overall understanding of this topic. I trust that I have been sufficiently clear in my argument. I am confident that you will be able to make minor adjustments to the description of this paragraph to ensure that the style is consistent with the academic standards without resorting to excessive emotional appeal.

2.3  Section 3.1. The first paragraph. Please edit this paragraph. The following information is duplicated in it. Description of the composition of sensitizers. Compare lines 145-152 and lines 154-158.

2.4  Section 3.2. The authors write: “This finding implies that linear molecular structures are correlated 239 with a longer conjugation length.” Maybe that's why a large number of papers describe the synthesis and study of linear structures? You may find fresh links with linear structures similar to the ones you synthesized. It would be interesting to make a brief comparison with the results of other similar studies.

2.5  Section 3.2. Line 262 and word “types”. in my opinion, it is not quite appropriate to talk about “types”, rather about “kinds”, as you wrote earlier.

2.6  Section 3.2. Reference [44] is not the correct reference. This paper does not deal with naphthadithiophene structure, but it has a reference to the correct article.

2.7  Section 3.4. The authors write: “Remarkably, the Voc values of shPS and tPS are noticeably higher than those of 336 typical metal-free organic sensitizers reported in the literature [47].” (336-337 lines). Firstly, the link [47] is erroneous. (Also, link [47] is incorrect in line 321.) It is more probable that reference [46] is intended. Secondly, there are examples of analogous structures in the literature for which the discussed values are higher than those presented in this study. Therefore, it is not entirely accurate to state that the compounds presented in this paper have such exceptional values. Based on the studies published to date, it can be confidently stated that the values discussed are higher than those of some examples of metal-free organic sensitizers. However, the order of values is approximately the same.

2.8  Section 3.4. The authors write:The origin of this behavior is still under investigation, but π-bridges of 9,9-dimethylflorene and naphtho[1,2-b:4,3-b']dithiophene are hypothesized to promote Voc in other types of donor systems.” This statement should be confirmed by reference, in my opinion.

2.9  Conclusions. The concluding sentence of this section on the subject of this work is excessively pretentious, given the volume of the preceding material.

Comments on the Quality of English Language

Line number

Typo

Amendment

41

potential in

potential for

3, 45, 48, 63, 67, 218, 249

photosensitizer

photosensitizers

55

within photovoltaic solar cell

within a photovoltaic solar cell

56

cost

the cost

57

photosensitizer-like

photosensitizer like

58

complexes-have

complexes have

63

metal free-organic

metal-free organic

63

a variable structure

a variable structures

63-65

Names of dyes

They don't need to be capitalized

66

is still on the

are still low

68

spectra and

spectra, and

87

and were

and introduced

90

and N,N-dimethyl

and N,N-dimethyl

91

nitrogen containing

nitrogen-containing

94

Ultraviolet

ultraviolet

95

(PMA) followed

(PMA), followed

98, 99

H, and 75 (125)

H and 75 (125)

101

constant(s)

constants

102, 103

on Mattson galaxy

on the Mattson Galaxy

106

spectrum were

spectra were

108

received unless

received, unless

117

of dye-

of a dye-

117

double layered-films

double-layered films

119

hrs

h

125,126

KLA-tencor

KLA-Tencor

129

Ethanol-

The ethanol-

136

PV measurement

PV Measurement

138

75 W

75W or 75 watt

139

generate monochromatic

generate a monochromatic

144

new four

four new

146,147

(4-(bis(9,9-dimethyl-9H-fluoren-2-yl)amino

4-bis(9,9-dimethyl-9H-fluoren-2-yl)amino

148

of current

of the current

153

are (information) shown

is shown

157

designed

the designed

158

as denoted

denoted as

163

loPS and cPS

loPS and cPS,

169, 215

metal free organic photosensitizer

metal-free organic photosensitizers

175

derivative

derivatives

198

cPS in

cPS, in

213

of tPS

of tPS

243

those

that

271

4-(bis(9,9-dimethyl-9H-fluoren-2-yl)amino

4-bis(9,9-dimethyl-9H-fluoren-2-yl)amino

274

rather

rather

331

,and open-

,and open-

344

. loPS

. LoPS

Author Response

  1. Abstract. Line 17. The word "unique" is not entirely appropriate in this context due to the fact that the properties of the compounds presented in this paper are not unique, but rather typical of them.

We thank reviewer for the valuable suggestion. As you suggested, we removed that word in manuscript.    

  1. Introduction. The last paragraph. The description of the research in this paragraph is presented in a somewhat artistic style with excessive pathos about the results of the work. This impression arises from the use of the words "enthralling", "fascinating narrative", "shedding light" "the merits and challenges", "illuminating the ongoing search". It is not my intention to belittle the significance of your work. However, it is important to note that only five compounds are described in this study. Given the vast amount of information published today, this is an important study. Nevertheless, it is but one of many works aimed at understanding the complexity of donor-acceptor systems and their properties. In other words, the structure-property relationship observed in a series of five compounds is unlikely to be a significant contributor to the overall understanding of this topic. I trust that I have been sufficiently clear in my argument. I am confident that you will be able to make minor adjustments to the description of this paragraph to ensure that the style is consistent with the academic standards without resorting to excessive emotional appeal.

We thank reviewer for the valuable suggestion. We have changed last paragraph regarding this issue.

  1. Section 3.1. The first paragraph. Please edit this paragraph. The following information is duplicated in it. Description of the composition of sensitizers. Compare lines 145-152 and lines 154-158.

We thank reviewer for the valuable suggestion. We have revised and removed those parts regarding this issue.

  1. Section 3.2. The authors write: “This finding implies that linear molecular structures are correlated 239 with a longer conjugation length.” Maybe that's why a large number of papers describe the synthesis and study of linear structures? You may find fresh links with linear structures similar to the ones you synthesized. It would be interesting to make a brief comparison with the results of other similar studies.

We thank reviewer for the valuable suggestion. As you suggested, there are many types of organic dyes including polymers and their molecular structure result in their solar cell performances under various conditions. For example, performances of solar cell depend on amount of adsorbed dye molecules for Jsc and passivation effect of dye molecule on the surface of TiO2 for Voc. In the current work, molecular structure of shPS highlights the significance of a well-developed conjugation network for improved solar cell performance. Due to the absence of a planar thiophene ring in shPS, its extinction coefficient is smaller than those of other organic photosensitizers, resulting in a decreased light absorption range.

  1. Section 3.2. Line 262 and word “types”. in my opinion, it is not quite appropriate to talk about “types”, rather about “kinds”, as you wrote earlier.

We thank reviewer for the valuable suggestion. We have changed it regarding this issue.

  1. Section 3.2. Reference [44] is not the correct reference. This paper does not deal with naphthadithiophene structure, but it has a reference to the correct article.

We thank reviewer for the valuable suggestion. As you pointed out, ref#44 doesn’t handle naphthodithiophene structure. Ref#44 reported that the relationship between π-conjugation network, the push-pull effect and conversion efficiency and it supports our observations.

  1. Section 3.4. The authors write: “Remarkably, the Voc values of shPS and tPS are noticeably higher than those of 336 typical metal-free organic sensitizers reported in the literature [47].” (336-337 lines). Firstly, the link [47] is erroneous. (Also, link [47] is incorrect in line 321.) It is more probable that reference [46] is intended. Secondly, there are examples of analogous structures in the literature for which the discussed values are higher than those presented in this study. Therefore, it is not entirely accurate to state that the compounds presented in this paper have such exceptional values. Based on the studies published to date, it can be confidently stated that the values discussed are higher than those of some examples of metal-free organic sensitizers. However, the order of values is approximately the same.

We thank reviewer for the valuable suggestion. As you pointed out, ref#47 in line 321 is changed to ref#46 regarding first asking. Regarding second asking, we agree that it may result in confusing and so, ref#47 in line 337 was removed.

  1. Section 3.4. The authors write: “The origin of this behavior is still under investigation, but π-bridges of 9,9-dimethylflorene and naphtho[1,2-b:4,3-b']dithiophene are hypothesized to promote Voc in other types of donor systems.” This statement should be confirmed by reference, in my opinion.

We thank reviewer for the valuable suggestion. As we understand, these kinds of photosensitizers are newly designed and we have tried to figure out their performances very deeply. As on-going work, we concluded that this sentence stands alone as it is.

  1. Conclusions. The concluding sentence of this section on the subject of this work is excessively pretentious, given the volume of the preceding material.

We thank reviewer for the valuable suggestion. As you pointed out, conclusion was revised properly.

Reviewer 3 Report

Comments and Suggestions for Authors

The paper by Kim et cols. describes the synthesis and photovoltaic characterization of four organic dyes to be used in solar cells. The work is not exciting and needs to be polished before an eventual publication. There are some suggestions below

- page 2, line 66: Despite their promising sides, challenging issues are…

- page 2, line 80: challenges of D-π-A organic photosensitizers…

- page 2, line 93: Reactions were monitored by thin layer chromatography (TLC) with 0.25 mm?

- page 3, lines 121-123: a verb is missing in “In the sealed cell, an electrolyte solution composed of 0.6 M of 1-methyl-3-propyl imidazolium iodide (PMII), 0.1 M LiI, 0.03 M I2, and 0.5 M of 4-tert-butylpyridine (TBP) in acetonitrile.”

- page 4, line 149: “The introduction of five distinct π-bridges” In figure 1 I only see 4 different bridges [figures (b)-(e)]

- page 4: sentences in lines 146-147 and 153-156 are a copy one of the other

- page 4, scheme 1: this is not an scheme, but a Chart. I would suppress figure (a), and draw R = [fluorene drawing] on one side or below the other figures. “Pink color (donor group) and blue color (acceptor group)” but in the text it has been previously stated that “α-cyanoacrylic acid as an electron acceptor” which is in red color.

- pages 4-5: this detailed explanation on the synthesis requires the presence of the corresponding scheme(s) in the same page, rather than in the SI.

- synthesis: it must be clearly stated which compounds are new and which ones have been previously described. This is not clear to me and is an essential point to assess the suitability of the paper. New compounds must be completely characterized (H- and C-NMR, IR, MS and, eventually, UV)

- page 5, lines 188 and 206: SN2 (N as subscript, 2 in regular size)

- page 5, line 193: indicate the yields for isolated 12 and 13, rather than the combined yield, which is irrelevant.

- SI: compounds 13 and 17 are the same, why 2 different numbers? Compounds 12 and 20 are the same, why 2 different numbers?

- SI: from Scheme S5 to Scheme S6 numbering jumps from compound 21 to compound 43, why?

- page 5, line 214: NMR spectroscopy, mass spectrometry

- page 5, Figure 1: I don´t understand how the semicolons (;) are used

- page 5, line 224: what is the meaning of “shoulder peaks”? I see bands and one shoulder

- page 6, line 226: these bands indicate a pi-pi* transition, but a CT?

- page 6, lines 228-232: I don´t understand the relationship between these sentences and the UV.

- page 7, Figure 2: I don´t understand how the semicolons (;) are used

- page 7, line 295: “electron density surfaces of HOMO and LUMO for frontier molecular orbitals” HOMO and LUMO are the frontier orbitals

- page 8, Figure 3: I don´t understand how the semicolons (;) are used

- page 8, lines 328-330: I think this paragraph shoulb be located at the beginning of the section (line 313)

- page 8, line 330: “Figure 3(b) displays” Table 2?

- page 9, Figure 4: I don´t understand how the semicolons (;) are used

- There is no need to repeat continuously “the metal-free organic photosensitizer”, “newly synthesized organic photosensitizers”, “four metal-free organic photosensitizers” lines 145, 169, 185, 202, 220, 292, 310-311, 352-353…

Comments on the Quality of English Language

The authors have problems with the use of semicolons (;)

Author Response

  1. - page 2, line 66: Despite their promising sides, challenging issues are…

- page 2, line 80: challenges of D-π-A organic photosensitizers…

- page 2, line 93: Reactions were monitored by thin layer chromatography (TLC) with 0.25 mm?

- page 3, lines 121-123: a verb is missing in “In the sealed cell, an electrolyte solution composed of 0.6 M of 1-methyl-3-propyl imidazolium iodide (PMII), 0.1 M LiI, 0.03 M I2, and 0.5 M of 4-tert-butylpyridine (TBP) in acetonitrile.”.

We thank reviewer for the valuable suggestion. As you suggested, we have revised those parts properly in manuscript.    

  1. - page 4, line 149: “The introduction of five distinct π-bridges” In figure 1 I only see 4 different bridges [figures (b)-(e)]

- page 4: sentences in lines 146-147 and 153-156 are a copy one of the other.

We thank reviewer for the valuable suggestion. As you suggested, we have revised those parts properly in manuscript.

  1. - page 4, scheme 1: this is not an scheme, but a Chart. I would suppress figure (a), and draw R = [fluorene drawing] on one side or below the other figures. “Pink color (donor group) and blue color (acceptor group)” but in the text it has been previously stated that “α-cyanoacrylic acid as an electron acceptor” which is in red color.

We thank reviewer for the valuable suggestion. As we know, it depends on recommendation of journal. Even molecular structure would belong Chart in some papers, but it should be Scheme and Figures. In this case, it would be OK to belong to Scheme. As you request, those parts were revised properly.

  1. - pages 4-5: this detailed explanation on the synthesis requires the presence of the corresponding scheme(s) in the same page, rather than in the SI.

- synthesis: it must be clearly stated which compounds are new and which ones have been previously described. This is not clear to me and is an essential point to assess the suitability of the paper. New compounds must be completely characterized (H- and C-NMR, IR, MS and, eventually, UV).

We thank reviewer for the valuable suggestion. Detailed information for synthesis requires large amount of space in main draft as you observed in SI. So, we concluded that current short brief would be ok in main draft. Even you requested, we could not include lots of data (H- and C-NMR, IR, MS and, eventually, UV). We hope you understand in this issue.

  1. - page 5, lines 188 and 206: SN2 (N as subscript, 2 in regular size)

- page 5, line 193: indicate the yields for isolated 12 and 13, rather than the combined yield, which is irrelevant.

We thank reviewer for the valuable suggestion. We have changed it regarding this issue.

  1. - SI: compounds 13 and 17 are the same, why 2 different numbers? Compounds 12 and 20 are the same, why 2 different numbers?

- SI: from Scheme S5 to Scheme S6 numbering jumps from compound 21 to compound 43, why?.

We thank reviewer for the valuable suggestion. Please understand that compound number and product number are temporal number to explain final product.

  1. - page 5, line 214: NMR spectroscopy, mass spectrometry

- page 5, Figure 1: I don´t understand how the semicolons (;) are used

- page 5, line 224: what is the meaning of “shoulder peaks”? I see bands and one shoulder

- page 6, line 226: these bands indicate a pi-pi* transition, but a CT?

- page 6, lines 228-232: I don´t understand the relationship between these sentences and the UV.

We thank reviewer for the valuable suggestion. Some of questions looks similar with previous ones and we have changed and revised those parts properly.

  1. - page 7, Figure 2: I don´t understand how the semicolons (;) are used

- page 7, line 295: “electron density surfaces of HOMO and LUMO for frontier molecular orbitals” HOMO and LUMO are the frontier orbitals

- page 8, Figure 3: I don´t understand how the semicolons (;) are used

- page 8, lines 328-330: I think this paragraph shoulb be located at the beginning of the section (line 313)

- page 8, line 330: “Figure 3(b) displays” Table 2?

- page 9, Figure 4: I don´t understand how the semicolons (;) are used

- There is no need to repeat continuously “the metal-free organic photosensitizer”, “newly synthesized organic photosensitizers”, “four metal-free organic photosensitizers” lines 145, 169, 185, 202, 220, 292, 310-311, 352-353….

We thank reviewer for the valuable suggestion. Some of questions looks similar with previous ones and we have changed and revised those parts properly.
